# Lysophosphatidic Acid and Hematopoiesis: From Microenvironmental Effects to Intracellular Signaling

**DOI:** 10.3390/ijms21062015

**Published:** 2020-03-16

**Authors:** Kuan-Hung Lin, Jui-Chung Chiang, Ya-Hsuan Ho, Chao-Ling Yao, Hsinyu Lee

**Affiliations:** 1Department of Life Science, National Taiwan University, Taipei 10617, Taiwan; kuanhunglin0621@gmail.com (K.-H.L.); linda1992252@gmail.com (J.-C.C.); 2Department of Radiation Oncology, University of Texas Southwestern Medical Center, Dallas, TX 75390, USA; 3Wellcome Trust-Medical Research Council Cambridge Stem Cell Institute and Department of Haematology, University of Cambridge, Cambridge CB2 0AW, UK; yhh29@medschl.cam.ac.uk; 4Department of Chemical Engineering and Materials Science, Yuan Ze University, Taoyuan 32003, Taiwan; yao@saturn.yzu.edu.tw; 5Department of Electrical Engineering, National Taiwan University, Taipei 10617, Taiwan; 6Angiogenesis Research Center, National Taiwan University, Taipei 10617, Taiwan; 7Research Center for Developmental Biology and Regenerative Medicine, National Taiwan University, Taipei 10617, Taiwan; 8Center for Biotechnology, National Taiwan University, Taipei 10617, Taiwan

**Keywords:** lysophosphatidic acid, lysophosphatidic acid receptors, hematopoiesis, microenvironment, transcription factor networks, anemia

## Abstract

Vertebrate hematopoiesis is a complex physiological process that is tightly regulated by intracellular signaling and extracellular microenvironment. In recent decades, breakthroughs in lineage-tracing technologies and lipidomics have revealed the existence of numerous lipid molecules in hematopoietic microenvironment. Lysophosphatidic acid (LPA), a bioactive phospholipid molecule, is one of the identified lipids that participates in hematopoiesis. LPA exhibits various physiological functions through activation of G-protein-coupled receptors. The functions of these LPARs have been widely studied in stem cells, while the roles of LPARs in hematopoietic stem cells have rarely been examined. Nonetheless, mounting evidence supports the importance of the LPA-LPAR axis in hematopoiesis. In this article, we have reviewed regulation of hematopoiesis in general and focused on the microenvironmental and intracellular effects of the LPA in hematopoiesis. Discoveries in these areas may be beneficial to our understanding of blood-related disorders, especially in the context of prevention and therapy for anemia.

## 1. Introduction

Hematopoiesis, a developmental process that gives rise to all cellular components of blood, is highly complicated and regulated by numerous factors in hematopoietic organs, including the bone marrow and spleen in adults as well as yolk sac and liver during vertebrate development. In adult vertebrates, hematopoietic stem cells (HSC) undergo a process of proliferation and terminal specification to generate all types of mature blood cells [1,2]. However, during embryonic hematopoiesis, blood cells are generated from non-hematopoietic sources. In vertebrates, hematopoiesis during embryonic development occurs in two waves: first, the short-lived “primitive wave” is characterized by the generation of primitive erythrocytes with embryonic globin expression, which are responsible for rapidly delivering oxygen to the entire embryo [3,4]. Subsequently, the “definitive wave” generates long-term hematopoietic stem cells (LH-HSCs), which can differentiate into multiple lineages of blood cells [5]. However, the cell populations from which the primitive and definitive waves originate remain to be definitely identified. Demonstrated through lineage tracing and time-lapse imaging, hematopoietic progenitors in the definitive wave are considered to be generated from endothelial cells [6,7]. This specialized cell population, which is termed the “hemogenic endothelium” (HE), gives rise to blood precursor cells through a process called endothelial-to-hematopoietic transition [8]. However, recent studies have supported the hypothesis that primitive hematopoiesis originates from precursors cells expressing endothelial markers, such as CD31 and FLK1 [9,10,11]. These precursor cells, which are neither a component of the mesoderm nor the HE, are termed “hemogenic angioblasts” [5]. Taken together, these findings make it clear that specific endothelial cells give rise to all blood cells, both primitive and definitive, through endothelial-to-hematopoietic transition during early embryonic development. However, how each endothelial cell is fated to generate a specific subset of blood cells remains elusive. It is important to clarify the regulatory mechanisms governing this specification, such as potential transcriptional regulation or microenvironmental influence.

Lysophosphatidic acid (LPA) is a small, bioactive glycerophospholipid that is derived from cell membrane phospholipids through autotaxin (ATX) metabolism [12]. It is expressed at low concentrations in all eukaryotic tissue types and at high concentrations in biological fluids, especially the serum and plasma [13]. LPA exhibits its various functions through the activation of multiple transmembrane LPA receptors (termed LPA_1–6_). All LPA receptors (LPARs) belong to the G-protein-coupled receptor (GPCR) family. GPCRs are localized on the cell surface, contain seven transmembrane regions, and mediate between extracellular cues and signal transduction through the recruitment of G-proteins. By activating different Gα subunits, LPA initiates its various cellular functions [12,14]. Through these activation mechanisms, LPA is involved in important functions in mediating cancer progression [15,16]. In recent years, LPA has also been reported to be involved in stem cell differentiation [17]. For example, LPARs LPA_1–3_ have been identified in mouse embryonic stem cells (ESCs) [18]. Moreover, LPA has been suggested to regulate pluripotency in human mesenchymal stem cells (MSCs) through the transcriptional co-factors YAP and TAZ [19,20]. Activation of LPA_1_ stimulated the downstream MEF/ROCK to regulate the hippo signaling in MSC cells. On the other hand, both LPA and ATX have been identified in the bone marrow, indicating that the ATX-LPA axis is active in this organ, where they may play important roles in hematopoiesis [21]. Remarkably, an LPAR agonist has been shown to promote terminal myeloid lineage differentiation of HSCs [22,23,24,25]. Hence, it has become clear that LPA and its receptor are pivotal regulators in hematopoiesis. Thus, in this article, we review the microenvironmental and intracellular effects of LPA during hematopoiesis and myeloid differentiation in particular.

## 2. Signaling Pathways of LPA and Its Receptors

### 2.1. LPA Generation

LPA is one of the most important bioactive lipids in vertebrates. It is composed of a long fatty acyl chain, a glycerol backbone, and a polar phosphate head group [26]. This growth factor is generated from both extracellular and intracellular metabolism. Lysophospholipid, which is catalyzed by ATX, is the major source of the extracellular LPA circulating in biological fluids [27,28,29] (Figure 1). Interestingly, expression of ATX has been identified in the bone marrow [21]. Furthermore, ATX is copiously expressed in CD34^+^ HSCs and FLT3-ITD^+^ acute myeloid leukemia (AML) cells. Overexpression of ATX significantly promotes T cell migration and proliferation in these cells [30]. Moreover, conditional overexpression of ATX promotes the differentiation of monocytes into macrophages in mice [31]. ATX is also involved in the regulation of the pluripotency transcriptional program, which is mediated by leukemia inhibitory factor (LIF) and bone morphogenetic proteins (BMP) in HSCs [32]. On the other hand, intracellular LPA is synthesized by three enzymes localized in the endoplasmic reticulum (ER) or the mitochondrial membrane: glycerophosphate acyltransferase (GPAT), phospholipase A (PLA), and acylglycerol kinase (AGK) [33,34,35]. Previous studies suggested that PLA2 activity is regulated by cytokines including IL-1, GM-CSF, and TNFα. The PLA2-dependent MAP kinase/ERK activation might be involved in hematopoiesis regulation [36,37]. These results suggested that both intracellular and extracellular LPA metabolism pathways participate in blood cell differentiation. On the other hand, LPA is substantially degraded by lipid phosphate phosphatase (LPP), a Mg^2+^-independent enzyme that is abundantly expressed on the surface of endothelial cells [38]. Evseenko et al. reported the expression of LPP in the murine bone marrow [21]. LPP may therefore participated as a regulator of LPA signaling in the bone marrow. Taken together, the identification of the LPA-generation and LPA-degradation enzymes in the hematopoietic microenvironment suggests the significance of the ATX-LPA axis in hematopoiesis.

### 2.2. LPA Receptors

G-protein-coupled LPARs are located on the cell membrane and contain seven transmembrane domains. Upon activation by extracellular LPA, these receptors induce downstream signal transduction to regulate various physiological functions [39]. To date, at least six membrane-bound LPARs have been identified (Figure 2); LPA_1–3_ were originally classified as members of the endothelial differentiation gene (EDG) family and share similar structures [14,40], whereas LPA_4–6_ belong to the P2Y receptor family [39,41]. Chun and his colleagues firstly cloned the cDNA of ventricular zone gene-1 (vzg-*1*), which is further reported as homolog of murine edg-2. Overexpression of edg-2 in neocortical cells promoted the LPA-dependent cell-shape change, which suggested that LPA is the ligand of this receptor [42]. LPA_1_ and LPA_2_ are widely expressed in most tissues and interact with G_i/o_, G_q/11_, G_12/13_, or Gs upon activation [43]. Activated G proteins recruit downstream secondary messengers, such as MAP kinase, Akt, Rho, and PI3 kinase pathways, which promote cell proliferation and cell migration [2]. LPA_3_ couples with G_q/11_ and G_i/o_ to regulate PLC activation and Ca^2+^ mobilization [43]. P2Y receptors belong to purinergic receptor family, which is demonstrated as a key regulators in nervous system [44]. LPA_4_ is expressed specifically in the pancreas, ovaries, and thymus, where it raises intracellular Ca^2+^ and cAMP levels through activation of G_q/11_ and G_12/13_ [45]. LPA_5_, which is expressed at low levels in multiple tissues and is highly expressed in kidney, interacts with G_q/11_ and G_12/13_ to increase intracellular cAMP levels [46]. LPA_6_, which has been reported to participate in hair follicle development [47], is the most recently defined member of the LPAR family. The specific functions of each receptor have been elucidated in studies using LPA receptor gene-knockout mice. LPA_1_-deficient mice exhibited reduced suckling, which was attributed to olfactory defects, and developmental abnormalities in the neurological system, whereas LPA_2_ knockout animals had no obvious phenotype [48]. In addition, LPA_3_-knockout mice exhibited delayed uterine implantation, altered embryo spacing, and reduced litter sizes [49]. No noticeable phenotype was associated with LPA_4_ deletion. However, embryonic fibroblasts from LPA_4_-knockout mice displayed hypersensitivity to LPA-induced cell migration [50]. LPA_5_^−/−^ mice showed no obvious phenotype [51], but recent results have suggested that LPA_5_ might prevents cancer progression and disease development by inhibiting cell motility and migration [52].

### 2.3. The Functions of LPA in Stem Cells

In recent years, accumulating studies have revealed the importance of LPA-LPAR signaling in stem cell biology. ATX-LPA axis is reported to regulate the pluripotency of ESCs through the activation of PLC/Ca^2+^ signaling and c-myc expression [18,20,32]. LPA receptors have been identified in HSCs, vascular stem cells, bone marrow stromal cells (BMSCs), neurospheres from neural stem cells (NSCs), and embryonic stem cells (ESCs) [53]. In addition, some studies have suggested a role of LPA in mesenchymal stem cells (MSCs): the ATX-LPA axis has been shown to regulate migration of MSCs [54,55] and hair follicle morphogenesis [56] through LPA_1–3_ signaling. Moreover, LPA has been reported to stimulate osteoblastic differentiation in TERT-overexpressing human MSCs through interplay between LPA_1_ and LPA_4_ [57]. Furthermore, LPA promotes proliferation and migration of adipose-derived MSCs via reactive oxygen species (ROS) and expression of ADAM12 [58,59]. On the other hand, lysophospholipids play important roles in neurogenesis and neural stem cell (NSC) differentiation. LPARs have been identified on the surface of oligodendrocyte progenitor cells [60,61] and hippocampus precursors [62,63]. In contrast, activation of LPA_1/3_ signaling inhibits differentiation of neural progenitor cells derived from human ESCs [64]. Furthermore, it has been reported that induced pluripotent stem cells (iPSC) express LPA_1–4_. Activation of LPA receptors and stimulation of the downstream Rho/Rock pathway promotes differentiation and proliferation in these cells [65]. Taken together, these findings suggest the involvement of the LPA-LPARs axis in stem cell differentiation, especially MSC and NSC development. However, the roles of LPA during the hematopoietic process remain elusive.

## 3. Regulation of Hematopoiesis

### 3.1. Initiation of Hematopoiesis

Hematopoiesis is temporally and spatially controlled by numerous factors during embryonic development in mice (Figure 3). Embryonic hematopoiesis takes place in several developmental niches that are rapidly altered by environmental signals in accordance with the highly dynamic conditions throughout embryonic development [66]. Before embryonic day (E) 10.5, blood islands in the mesoderm of the extraembryonic yolk sac are the major site of blood cell generation [67]. During this period, hemogenic angioblasts, which are bipotent precursors of endothelial and hematopoietic cells, are the main source for primitive hematopoiesis [5]. Based on observations from in vivo single-cell differentiation [68] and in vitro ESC culture, mesoderm cells have also been found to generate HE to support definitive hematopoiesis in the yolk-sac [9]. In the later stage of murine embryonic development (around E10.5), hematopoiesis transitions to the embryo proper, in organs such as the aorta-gonad-mesonephros (AGMs) [69,70], placenta [71,72] and arteries [73]. In this stage, HSCs emerge from HE through transcriptional regulation by RUNX1 [74,75]. At around E11.5 of embryonic development, hematopoiesis migrates to the fetal liver, which is the major site of blood cell expansion and maturation in the embryo [66]. Finally, at around E16.5, blood cell development transitions from the fetal liver to the bone marrow where it forms colonies that form and maintain hematopoietic niches throughout adult life [66]. 

HSCs, which have the capability for self-renewal and long-term differentiation, are the source of support for all types of blood cells throughout the entire lifespan of adult vertebrates. HSCs generate two main cell lineages in vertebrates: the myeloid lineage (leukocytes, erythrocytes, platelets and dendritic cells) and the lymphoid lineage (T cells, B cells and NK cells). Due to the short lifespan of these mature blood cells, the self-renewal ability of HSCs is critical [76]. In order to maintain blood homeostasis, it is necessary to continuously produce a huge quantity of progeny to replenish the depleting pool of mature blood cells. Following division of an HSC in the niche, the two daughter cells can either remain as stem cells or progress to differentiation. This specialized process simultaneously maintains the stem cell pool size and produces an adequate number of mature blood cells in circulation [77]. The balance between self-renewal and differentiation thus needs to be tightly controlled. HSCs have extensive clinical utility due to their specialized characteristics and functions, particularly with regard to stem cell transplantation. Therefore, investigating the mechanisms underlying HSC differentiation is important for efficient clinical application of HSCs. In this study, we review the two main factors that affect HSC differentiation and physiology: microenvironments and transcriptional factor networks. 

### 3.2. Microenvironmental Factors

The physiological functions of HSCs are tightly controlled by complex structures in the bone marrow microenvironment. Such structures are called the “niche”, which was first defined by Schofield in 1978 [78]. Niche cells provide fundamental support for HSCs to regulate their quiescence, self-renewal and differentiation [79]. Disruption of niche-derived signals causes severe defects in HSC development. For example, ablation of osteoblast inhibited the expression of IL-7 and SDF-1, which are necessary for B-cell development [80,81]. In the bone marrow microenvironment, stromal cells are crucial in the regulation of HSC differentiation. They support HSC homeostasis by way of three functionalities [82]. First, stromal cells provide a suitable microenvironment for HSC differentiation. Furthermore, the extracellular matrix (ECM), which is secreted by stromal cells, supports the growth of HSCs in their specific niches. ECM plays a vital role in stem cell niches, where it serves as a seedbed for various cells in the environment, controlling and regulating pH, osmotic pressure, and providing mechanical and biochemical factors [82]. Secondly, stromal cells secrete cytokines and growth factors [82]. These paracrine factors drive HSCs toward different cell lineages. When co-cultured with stromal cells, purified CD34^+^ HSCs showed significantly higher expansion under cytokine treatment [83]. Various cytokines secreted by stromal cells (such as interleukin family [84], fibroblast growth factors [85], thrombopoietin [86,87], and stem cell factor “SCF” [88]) are involved in the regulation of early and late hematopoiesis. Upon activation of their receptors, downstream signaling leads to changes in gene expression and commitment of an HSC towards specific lineages. Finally, cell-cell interactions between stromal cells and HSCs also critically affect this differential regulation [89]. Basically, the niches in bone marrow are classified into two categories: the perivascular niche and the endosteal niche. 

#### 3.2.1. Perivascular Niche

The bone marrow is a highly vascularized organ; it has numerous arteries and veins that enter the tissue to create a complex structure. The vast majority of HSCs are located close to blood vessels in this microenvironment [90]. The HSCs that arose from the HE during embryonic development remain associated with blood vessels, suggesting a juxtacrine interaction between these cells [91]. In this niche, endothelial cells (ECs) play a critical role in regulating biological processes in HSCs. They produce indispensable factors, such as SCF, Notch ligands, and CXCL12, that are responsible for HSC self-renewal and differentiation [91]. Furthermore, the stromal cells that express leptin receptor (LepR) and nestin are the major source of CXCL12 and SCF, which maintain HSC in the junction between sinusoids and arteries [92,93]. On the other hand, neural components that associate with arteries also provide signals for HSC homeostasis in the bone marrow. Sympathetic neurons release norepinephrine to control CXCL12 production, which in turn affects HSC trafficking from bone marrow to blood [94]. Moreover, megakaryocytes produce TGF-β and thrombopoietin (TPO) to restrict the proliferation of HSCs [95,96]. This mechanism, in which HSCs are regulated by their own descendants, suggests the existence of a feedback loop during hematopoiesis. 

#### 3.2.2. Endosteal Niche

Imaging studies and fractionation results have shown that HSC numbers are enriched and associated with the endosteal (inner) surface of the bone marrow, supporting the concept of the endosteal niche [97,98,99]. Bone-lining osteolineage cells secret HSC regulatory molecules, such as embigin and angiogenin, to regulate the quiescence of HSCs [98,100]. Previous studies have shown that constitutive expansion of osteoblasts leads to increased numbers of HSCs, which strongly suggests the importance of osteoblasts in the endosteal niche [101]. In addition, osteocytes have been reported to regulate the expansion of myeloid progenitors and G-SCF-induced mobilization of HSCs [102]. On the other hand, some studies have suggested contrary roles of osteoclasts in HSC regulation, such as proliferation and mobilization of HSCs [89,103]. Macrophages are another type of HSC progeny cell that regulates HSC function. Studies have suggested that macrophages that are enriched adjacent to the bone surface, referred to as “osteomacs”, restrict HSCs around the endosteal niche by providing CXCL12-dependent adhesion [89,104,105]. 

### 3.3. Transcriptional Factor Networks

Upon induction by extracellular signals, such as cytokines or cell-cell contact, intracellular signals evoke the functions of transcription factors (TFs) to regulate gene expression, which is responsible for determination of cell fate in HSCs. To date, hundreds of TFs involved in the regulation of hematopoiesis have been identified, making it difficult to delineate their interactions. However, the advancement of technologies in genome-wide research and computational science has given rise to accurate network models. In this review, we describe some TF interactions that have been well-recognized in hematopoiesis.

#### 3.3.1. EKLF-FLI1 Interactions

Erythroid Krüppel-like factor (EKLF) and Friend leukemia integration 1 (FLI1) are two well-studied TFs involved in the regulation of HSCs during terminal differentiation towards the myeloid lineage. EKLF promotes erythropoiesis [106,107], whereas FLI1 activates gene expression during megakaryopoiesis [108]. EKLF suppresses FLI1–dependent transcription of the GPIX gene during megakaryopoiesis, and EKLF-dependent transcription of the β-globin gene is ablated in MEF cells transfected with FLI1 [109]. Previous findings have also suggested that EKLF and FLI1 have physical contact and suppress the activity of each other [109]. EKLF and FLI1 might also recruit cofactors, such as GATA-1, while forming multiprotein complexes that selectively activate the promoters of either erythrocytic or megakaryocytic genes [110].

#### 3.3.2. GATA Switch

Discovery of a DNA sequence that commonly resides in the promoter region of globin provided the evidences for the identification of GATA motif (WGATAR). This observation led to the identification of the protein family binding to this sequence: the GATA family of transcription factors [111,112]. GATA-1, -2, and -3 are classified as hematopoietic GATA factors based on their important activities during HSC differentiation [113], whereas GATA-4, -5, and -6 play critical roles in cardiac function and lung tissue [114]. GATA-1 is preferentially expressed in erythroid or megakaryocytic progenitors, while GATA-2 is abundantly expressed in HSCs and myeloid progenitors [112,113,115]. During hematopoiesis, the balance between the expression of GATA-1 and GATA-2 acts as a developmental driver to promote myeloid differentiation [116]. In progenitor cells, GATA-2 positively regulates its own expression by directly binding to the promoter region of the GATA-2 and p38/ERK/CXCL2 circuits [117,118]. Chromatin immunoprecipitation (ChIP) has shown that GATA-2 occupies the promoter region of GATA-1. Upon stimulation by cytokines required for myeloid differentiation, release of TAL and LSD1 from the promoter of GATA-1 leads to the expression of GATA-1 [119]. Furthermore, ChIP analysis has indicated that GATA-1 binds to the upstream region of the -70-kb GATA-2 promoter domain, and thus represses the expression of GATA-2 with FOG-1 occupancy [120]. This “GATA switch model”, in which GATA-1 displaces GATA-2 during HSC differentiation, induces a subset of transcription output and leads to lineage commitment [121]. 

#### 3.3.3. PU.1/GATA-1 Antagonism

The interaction between PU.1 and GATA1 is an another example of the determination of lineage commitment by TFs. PU.1 is responsible for monocytic differentiation in the myeloid lineage, while GATA-1 induces erythropoiesis and megakaryopoiesis [122]. Furthermore, multiple lines of evidence have indicated that GATA-1 reduces the effects of PU.1 by interacting with its regulatory regions to suppress its expression [123,124]. Mathematical modeling has shown that this mutual repression also inhibits the opposing cis-transactivators or downstream target genes, further results in lineage commitment [125].

## 4. Microenvironmental and Intracellular Effects of LPA during Hematopoiesis

Hematopoiesis is tightly controlled by the microenvironment surrounding HSCs. As previously described, cytokines, ECM, and stromal cells play major roles in their niche. Since the bone marrow is the main hematopoietic organ in adult humans, we here review the functions of LPA in bone marrow. Recent studies have indicated that autotaxin and lipid phosphate phosphatases 2A (PPAP2A, the enzyme that degrades LPA) are differentially expressed in bone marrow [21], suggesting the presence of LPA. In recent years, an increasing number of studies have supported the concept that LPA “indirectly” affects hematopoiesis by participating in the regulation of osteogenesis and stromal cell activity. LPA has been reported to regulate osteogenesis by promoting the survival and proliferation of osteoblasts [126,127]. Furthermore, the MLO-Y4 osteolineage cell responds to LPA by triggering dendrite outgrowth [128]. Moreover, blockade of LPA_1_ signaling reduced the number of newly formed osteoclasts [129]. On the other hand, Kanehira et al. showed that pharmacological activation of LPA_1_ delays culture-dependent senescence of human marrow stromal cells [130]. In addition, the LPA-LPA_4_ axis regulates hematopoiesis indirectly by affecting stromal cell activity to support hematopoiesis in the bone marrow [131]. Since osteolineage cells have been reported to regulate the mobility and activity of HSCs [98,100], these lines of evidence suggest the importance of the LPA-LPAR axis in the regulation of hematopoiesis. 

LPA and its analogue, S1P, have been suggested to stimulate proliferation and mobility of CD34^+^ HSCs [30,132,133,134]. S1P and LPA were also reported to promote the ability of primitive hematopoietic cells to invade into the stromal cell layer [135]. During early development, the ATX-LPA axis has been reported to participate in the regulation of hemangioblast differentiation through activation of LPA_1_-induced phosphoinositide 3-kinase/Akt and Smad signaling [136]. These results support the “direct” regulation of LPA during early hematopoiesis in the embryo. On the other hand, there is increasing evidence to suggest that LPA mediates HSC lineage commitment, especially towards the myeloid lineage but not the lymphoid lineage [17]. Moreover, gene profiling results have confirmed the expression of LPARs in myeloid progenitors [21]. LPA_1_ is also involved in NM23-dependent myeloid differentiation of leukemia cells [137]. On the other hand, LPA activates the Akt-mTor-PPARγ signaling axis to convert monocytes into macrophages [31]. Taken together, these results suggest novel and critical roles of LPA signaling during early hematopoiesis in the embryo and HSC differentiation in the adult. Nevertheless, the roles of LPA and LPA receptor in terminal differentiation, such as erythropoiesis and megakaryopoiesis, have not yet been fully confirmed.

Our laboratory investigated the functional roles of LPA in myeloid differentiation of HSCs. We found that knockdown of LPA_3_ in zebrafish embryos led to severe erythrocyte defects [23], suggesting that this particular LPA receptor is required for red blood cell (RBC) development during early hematopoiesis. Our results also suggested that the LPA-LPA_3_ axis is involved in erythropoietin (EPO)-dependent erythropoiesis and nuclear translocation of β-catenin. In addition, the effects of LPA on erythropoiesis were diminished by blocking c-Jun activated kinase and PI3K/AKT, which are the downstream signals of EPO receptor. These results suggested that LPA might have a synergistic role with EPO in the regulation of erythropoiesis [23]. We further reported that activation of LPA_2_ and LPA_3_ oppositely regulated erythropoiesis [22]. In that study, we demonstrated Stimulation via LPA_2_ and LPA_3_ agonist exerts antagonizing effects on erythropoiesis in both K562 cells and CD34^+^ HSCs. Furthermore, exposure to LPA_3_ agonist 2S-OMPT rescued the erythropoietic defect caused by zLPAR3 morpholino oligonucleotide injection in zebrafish. Treatment of embryos with LPA_2_ agonist GRI977143 (GRI) resulted in an increased percentage of embryos with severe RBC defects. Finally, our results showed that the RBC count, hemoglobin (HGB), and hematocrit (HCT) decreased significantly after GRI injection, and conversely increased after 2S-OMPT treatment in BALB/c mice. Overall, our results suggest that agents similar to GRI and 2S-OMPT could serve as powerful drug candidates to regulate erythropoiesis. We also demonstrated the inhibitory role of LPA_2_ during RBC differentiation. Our group also found that LPA_3_ affects differentiation in megakaryopoiesis [25]. Notably, these results from pharmacological activators and shRNA experiments have demonstrated the negative regulatory role of LPA_3_ during megakaryocyte differentiation in CD34^+^ HSCs. Furthermore, morpholino-mediated blockage of LPA_3_ translation led to increased numbers of CD41-GFP^+^ cells in Tg(CD41:eGFP) zebrafish. Taken together, our results suggest that the balance between the expression levels of LPA_2_ and LPA_3_ might be a molecular switch for cell fate determination in the myeloid lineage (Table 1).

## 5. LPA Receptor Agonists and Antagonists as Potential Treatments for Anemia

Anemia is a disease featuring a decline in the numbers of RBCs and a reduction of HGB or HCT below normal range. A variety of medications have been developed to treat this disease. To date, EPO is the most popular medicine to treat anemia caused by chronic kidney disease (CDK), cancer, or AIDS. For example, Epogen, the most popular medication, has been developed for over 28 years for clinical use in patients with anemia [138]. The main component of Epogen is recombinant erythropoietin (Epotein-alfa), which is produced in Chinese hamster ovary (CHO) cells. Recent studies have focused on biosimilars of erythropoietin. Last April, Pfizer’s biosimilar medicine Retacrit was approved by the US-FDA. Related products, such as Recormon (Epotein-beta) [139] and Aranesp (darbepoetin alfa) [140] are commonly used for patients with anemia. However, recent studies have reported that overdose of EPO treatment may cause many side effects, including cardiovascular disease and tumorigenesis [141]. On the other hand, EPO is a protein pharmaceutical. Its high cost of production may constitute a substantial burden for health insurance agencies in many countries. Therefore, it is extremely urgent to development a new medication, preferably small-molecule drugs, to replace EPO. Inhibitors of the hypoxia-inducible factor (HIF) prolyl hydroxylase (PH) enzyme are a new class of agents for the treatment of anemia [142]. HIF is hydroxylated by HIF-PH to cause its degradation [143]. On the other hand, HIF is reported to play important roles in cell-cycle regulation of HSC and production of EPO. Therefore, prevention of HIF degradation by HIF-PH inhibitors might be a novel target for anemia treatment [144].Vadadustat (Akebia Therapeutics, Cambridge, MA, USA), Roxadustat (FibroGen, San Francisco, CA, USA) and Daprodustat (GlaxoSmithKline, London, UK) are recently developed HIF-PH inhibitors currently undergoing clinical trials [144,145]. 

Most of the LPA agonists and antagonists are currently being tested in preclinical trials for therapeutics for cancer or idiopathic pulmonary fibrosis [146]. We were the first group to show that the activation of LPA_3_ by its agonist, 2S-OMPT, strongly increased the numbers of RBCs in zebrafish embryos and adult mice. Furthermore, our unpublished results show that LPA_3_ agonists hold potential as therapeutics for anemia. We used phenylhydrazine to induce hemolytic anemia in a mouse model. Administration of 2S-OMPT not only rescued the reduction in erythrocytes, but also the HGB and HCT levels in the peripheral blood. It also increased the athletic ability of mice, as analyzed by voluntary wheel running. We suggest that a new LPA_3_ agonist should be developed as a potential treatment for anemia, since such a small-molecule agent would have lower cost and higher specificity for its target. 

## 6. Conclusions

The hematological roles of LPA is increasingly apparent during blood cell development and disease. Previous investigations have implied that LPA plays key roles in the regulation of early stage of progenitor cells through their support of the precise regulation of HSC niches. Our results led to an important conclusion: LPA not only affects immature precursors, but also provides crucial signals during the terminal stages of myeloid lineage specification. Moreover, given that myeloid progenitors express LPA receptors in different patterns and combinations, it is likely that their effects are precisely balanced with the regulation of stem cell fates. Since LPA_3_ agonists have momentous effects on RBC numbers and behavior in mice, our studies may provide a new therapeutic strategy for anemia.

## Figures and Tables

**Figure 1 ijms-21-02015-f001:**
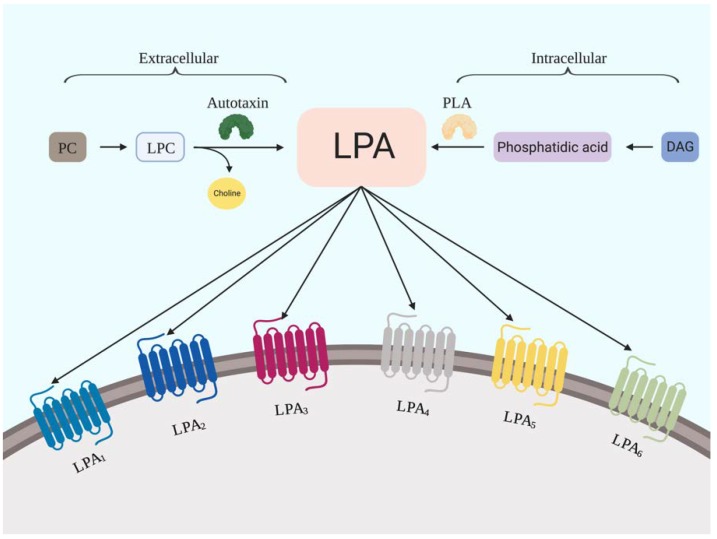
LPA metabolism. The generation of LPA is catalyzed by extracellular autotaxin or intracellular phospholipase A (PLA).

**Figure 2 ijms-21-02015-f002:**
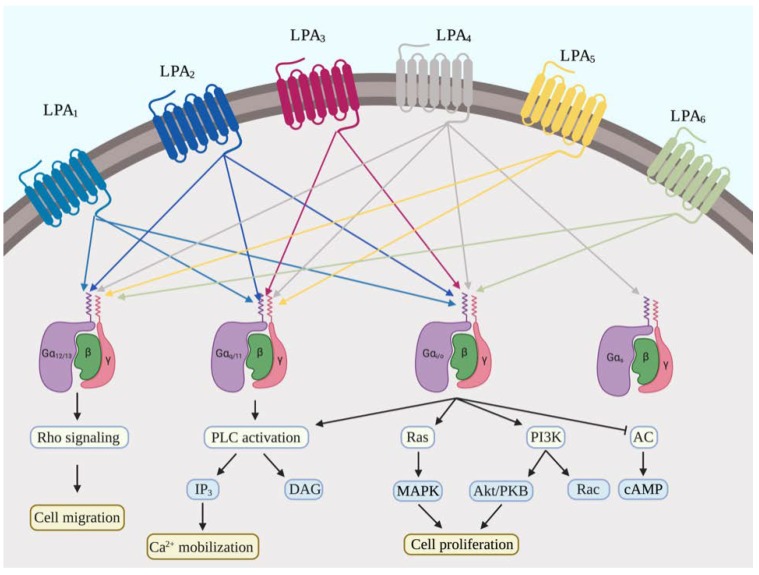
LPA receptor signaling. The functions of LPA are induced by the activation of 6 G-protein-coupled LPA receptors.

**Figure 3 ijms-21-02015-f003:**
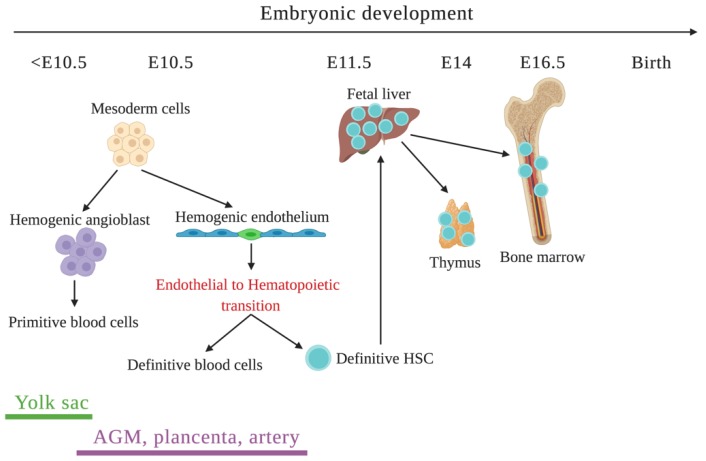
Hematopoiesis during murine embryonic development.

**Table 1 ijms-21-02015-t001:** Microenvironmental and intracellular effects of LPA-LPAR axis during hematopoiesis. BM MSCs: bone marrow mesenchymal stem cells; HSPC: hematopoietic stem and progenitor cells; MEPs: megakaryocyte-erythroid progenitors.

	Cell Type	LPA-Related Effects	Reference
Extracellular/ Indirect effects	BM MSCs	Detectable level of ATX expression	[21]
	BM MSCs	Activation of LPA_1_ retards cell senescence	[129]
	Endosteal osteoblasts	High PPAP2A expression level	[21]
	Endosteal osteoblasts	LPA promotes survival and proliferation	[121,126]
	MLO-Y4 cell line	LPA triggers dendrite outgrowth	[127]
	BM PDGFRα^+^ cells	LPA_4_ promotes the production of HSPC proliferation factors	[130]
Intracellular/ Direct regulation	THS119 cell line	LPA-LPA_1_ axis promotes invasion ability	[134]
	Hemangioblasts	LPA_1_ activates hematopoietic differentiation	[135]
	Leukemic cell lines	LPA_1_ is involved in NM23-dependent myeloid differentiation	[136]
	Monocytes	Activation of LPA-Akt-mTor-PPARγ signaling converts monocytes into macrophages	[31]
	CD34+ HSPCs	LPA promotes the early stage of myeloid differentiation	[21]
	MEPs, K562 cell line	Activation of LPA_3_ promotes erythrocyte differentiation	[23,137]
	MEPs	LPA-LPA_3_ axis inhibits megakaryocyte differentiation	[25]
	MEPs, K562 cell line	LPA_2_ inhibits both erythropoiesis and megakaryopoiesis	[24,137]

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
