# Peer review of "Lysophosphatidic Acid and Hematopoiesis: From Microenvironmental Effects to Intracellular Signaling"

_ijms, 2020, doi:10.3390/ijms21062015_

Round 1

Reviewer 1 Report

This is an interesting and useful review about roles for LPA signaling in control of hematopoiesis.   Some issues that are omitted or could be discussed in more detail include:

  1. Although autotaxin probably generates most circulating LPA there are other potential and proven pathways for generation of extracellular LPA that need to be discussed here.  This includes the phospholipase A mediated pathways described by Aoki and colleagues.
  2. Similarly, there is a significant body of evidence identifying integral membrane lipid phosphate phosphatases as regulators of LPA metabolism and signaling that likely impact on these processes. This issue needs to be discussed.
  3. The LPA related bioactive lipid mediator sphingosine 1 phosphate has roles in hematopoiesis that, in particular, involve trafficking of cells between the bone marrow and the systemic circulation. Some discussion of these issues and how they relate to the postulated roles for LPA in hematopoiesis also needs to be included here.

Author Response

Reply to Q1:

Thanks for your suggestion. Descriptions about the importance of PLA in hematopoiesis were added (L91~L95). Please refer to the end of page 2.

Reply to Q2:

Descriptions regarding the identification of LPP in bone marrow were added between L95~L108, which are marked by red color. Please refer to the beginning of page 3.

Reply to Q3: 

Thanks for your suggestion. S1P is indeed an important molecule and is currently reported as a hematopoietic regulator.The functions of LPA and S1P were described between L335-L337, which are marked by red color. Lidgerwood et al. have excellently elaborated the function of S1P in stem cell biology therefore we did not provide further descriptions on this issue.

Reviewer 2 Report

Lin et al. conceived a concise review on the roles of LPA on hematopoiesis, partially I have the impression that a third figure would help in understanding the complex regulatory mechanisms, especially for the niche differentiations I would find such a figure very useful. Maybe a second table on that topic could also help to get a better overview on this. A major redesign of figure 1 is necessary to greatly improve the visualization.  Another general comment: all citations are missing a space character beforehand. Detailed suggestions are following:

  • L21: exhibits instead of generates..
  • L59/60: rephrase please
  • Figure 1: I would suggest splitting this figure into two panels for a) structure (I miss this throughout the whole paper) and generation of LPA and b) for the functions of LPA
    • Label the LPARs on top in black and do not use differential colors for the names.
    • Rethink the colors of the LPARs? LPA4 and LPA6 are hard to distinguish.
    • The interaction lines are too "crowded", maybe using dashed lines could improve the visuals?
  • L111: A little bit more details on the EDG and P2Y families would help here.
  • Figure 2: Here I had severe problems with the quality of this figure in the reviewer PDF... maybe a review system artifact?
    • Please use smaller font and arrow size for the topline Embryonic dev...
    • The arrowheads could be synchronized to have the same shape - in the current state the different arrowheads are mildly irritating.
    • At the bottom line: Yolk sac, AGM... etc -> why not indicate this with colored bar lines instead of arrows?
  • L194: Rephrase
  • L195ff: why is this sentence written in bold?
  • L228: ...studies...
  • L303: Citation is in a smaller fontsize...
  • L344ff: A mixture chaos of different fontsizes irritates me a lot here. Correct this please.
  • L352: HIF-PH needs a bit more introduction or explanatory sentences here.

Author Response

Reply to the suggestions in L21: Thanks for your notification. It has been corrected.

Reply to the suggestions in59/60: Thanks for your suggestion. It has been rephrased (in L60).

Reply to the suggestions in L111: Please find the added description about EDG and P2Y family between L121~129.

Reply to the suggestions in L194: Thanks for your suggestion. It has been rephrased (in L227).

Reply to errors in L228: Thanks for your notification.It has been corrected.

Reply to errors in L261: Thanks for your notification. It has been corrected.

Reply to errors in L303: Thanks for your notification. It has been corrected.

Reply to errors in L344: Thanks for your notification. It has been corrected.

Reply to the suggestions in L352: Please find the added description for the relationships of HIF and HIF-PH between L389-L391.

Reply to suggestions for Figure 1 and Figure 2: Figure 1 was split into new Figure 1 and new Figure 2. Font size and arrowheads are also synchronized.